# Modeling Environmentally Conscious Purchase Behavior: Examining the Role of Ethical Obligation and Green Self-Identity

Rakesh Kumar [1] , Kishore Kumar [2] , Rubee Singh [3] , José Carlos Sá [4] , Sandro Carvalho [5] and Gilberto Santos [6,*]

1 School of Management Studies, Motilal Nehru National Institute of Technology Allahabad, Prayagraj 211004, India; rakesh@mnnit.ac.in
2 School of Business and Management, Christ (Deemed to be University) NCR Campus, Ghaziabad 201003, India; akishore001@gmail.com
3 Institute of Business Management, GLA University, Mathura 281406, India
4 School of Engineering (ISEP), Polytechnic of Porto, Rua Dr. António Bernardino de Almeida 431, 4200-072 Porto, Portugal; cvs@isep.ipp.pt
5 Technological School, 2Ai—Applied Artificial Intelligence Laboratory Polytechnic Institute of Cavado and Ave, 4750-810 Barcelos, Portugal; scarvalho@ipca.pt
6 Design School, Polytechnic Institute Cavado Ave, Campus do IPCA, 4750-810 Barcelos, Portugal
* Correspondence: gsantos@ipca.pt

**Abstract:** Due to environmental degradation, using environment-friendly products has become necessary to reduce carbon emissions. However, the consumption of such products is still below expectations because these products are usually costlier than their traditional counterparts. The current study aims to investigate consumer behavior towards environment-friendly products using Ajzen's theory of planned behavior as a theoretical model. The study seeks to examine the role of the key determinates of environmentally conscious purchase behavior, such as ethical obligation and green self-identity. A total of 386 responses were collected from consumers living in a few major cities of northern India using purposive sampling. The data were analyzed using structural equation modeling in Amos 22.0. The results demonstrated that attitudes towards environment-friendly products perceived behavioral control and green self-identity as the major determinants of green purchase intentions. In addition, attitude was reported to mediate the effect of ethical obligation on green purchase intentions and green self-identity was found to moderate the effect of attitude on green purchase intentions. Additionally, green self-identity was also reported to moderate the relationship between ethical obligation and attitude. The study adds value to the existing literature by signifying the role of green self-identity and ethical obligation in stimulating consumers' green purchase intentions. The findings of the study are also meaningful for marketers and policymakers.

**Keywords:** ethical obligation; green self-identity; theory of planned behavior; green purchase behavior; green products; environment-friendly products

## 1. Introduction

Global warming and climate change are the most dangerous threats to the very existence of human beings. They have affected almost all walks of human life, including the consumption of various products and services. The excessive emission of carbon dioxide has caused global warming and resulted in climate change, which are influencing the ecosystem of the earth [1,2]. According to a report of the Inter-Governmental Panel for Climate Change, approximately 1 °C of global warming above pre-industrial levels has been witnessed due to human activities, which is expected to reach 1.5 °C between 2032 and 2052 [3]. The report focuses on the urgent and unprecedented actions required to check the rate of carbon emission as the world has already witnessed the severe consequences

of 1 °C of global warming in the form of 'rising sea levels', 'diminishing Arctic Sea ice', 'extreme weather', etc.

Carbon emission at the global level has been on a continuous rise, despite the pandemic, when in many countries, industrial and other activities were very much restricted [4]. The adverse effects of rising temperatures and global warming would have to be faced by poor and vulnerable people, especially in developing countries. Therefore, it has become important to reduce the use of those products that are harmful to the environment. Thus, the promotion of green practices and environmental activities is necessary [5]. This requires an urgent shift from traditional practices to green and sustainable practices, including the consumption of goods and services [6]. As the issue of global warming is attracting global attention, even marketers are now shifting from traditional marketing practices to the marketing of more sustainable and green products, also called eco-friendly, environment-friendly products, or green products [7]. Moreover, adopting sustainable business practices improves firms' performance [8] and helps in creating value for the company [9]. In addition, becoming environmentally sustainable in terms of business practices provides such companies an edge over their competitors [10], which attracts other companies to adopt green business practices.

The general awareness and concern about environmental issues among consumers have increased in the last couple of decades [11]. Environmentally conscious consumers are inclined towards sustainability and show strong intentions to adopt sustainable consumption. These consumers are now looking for sustainable and environment-friendly products [12]. Thus, it has become important to investigate the major determinants of green purchase behavior. Ajzen's [13] "Theory of planned behavior" (or TPB model) is a very popular theoretical model and has frequently been used to predict an individual's purchase behavior, including green purchase behavior [12–19]. However, green consumption has been reported to be influenced by a number of sociocultural, demographic, and psychological factors [20–23]. Moreover, psychological factors, such as ethical obligation [24] and green self-identity [25], have been found to be critical predictors of green purchase intention. Issues concerning ethics, moral beliefs, and values have always attracted researchers across the globe to explore new dimensions in understanding the role of ethics and moral values in individual behavior [13].

The shift of consumption habits from traditional products to green products, which are relatively expensive, is the result of the ethical decision-making process, which is guided by moral principles and values. Ethical obligation refers to the extent to which an individual feels an obligation towards others [26]. Therefore, it becomes interesting to investigate how a person's tendency to comply with ethical principles influences their intention to purchase green products. Additionally, environmentally conscious consumers are found to exhibit favorable intentions towards green products as a notion of the manifestation of their responsibility towards the environment [27]. They engage in sustainable consumption to portray their distinct image in society as a sensible individual. Green self-identity is the relatively recent adaptation of the self-identity concept in the domain of sustainability, while ethical obligation is relatively less explored. In addition, both ethical obligation and green self-identity have been investigated separately as antecedents of green purchase intention; however, the current study examines the connection between the two constructs using Ajzen's TPB model, which is one of the most widely used theoretical models in the domain of sustainable consumption. However, Ajzen [13] himself advocated to extend this model with context-specific variables, such as socio-cultural, demographic factors, while extending it in other contexts. Therefore, this study extends the TPB model to ethical obligation and green self-identity in the present context.

Therefore, the current study proposes to examine the effect of ethical obligation and green self-identity in predicting the purchase intentions of young and educated Indian middle-class consumers (living in northern India) towards sustainable products. The study aims to fill the above-mentioned gap and focuses on the following objectives. First, the study proposes to add ethical obligation and green self-identity to the basic structure of the

TPB model, while extending this model to predicting green consumption intentions. Second, the study seeks to investigate both the direct and indirect effects of ethical obligation on green purchase intentions. Third, the study proposes to explore the moderating role of green self-identity in green consumption behavior. Green self-identity has mostly been considered as a direct predictor of purchase intention; however, the moderating role of green self-identity is rarely explored. The investigation of the moderating effect of green self-identity may provide some valuable insights and may add value to the extant literature.

The current manuscript has been divided into seven sections. The first section (introduction section) details the background of the research problem undertaken in the current study, along with the research gap and objectives of the study. The second section provides the theoretical framework and development of the hypotheses for the study. The third section indicates the material and methods used in the study, while the analysis and results are presented in the fourth section. Moreover, these results are discussed and synthesized in the fifth section, while the sixth and seventh sections present the implications and conclusions, respectively.

## 2. Theoretical Framework and Hypotheses

The TPB model is an extension of the "Theory of reasoned action" (or TRA model) given by Fishbein and Ajzen [28], which postulated that behavioral intentions may be predicted by an individual's attitude towards the behavior and subjective norms. Attitude is a "learned predisposition to behave in a consistently favorable or unfavorable way with respect to a given object" [29] (p. 258). Thus, attitudes are the outcome of an individual's evaluation of an object, good, or service. Subjective norms (SNs), on the other hand, are an individual's tendency to comply with social norms and expectations [28]. Attitude and SNs have been assumed to be the most important predictors of behavioral intention. However, the extent to which these two predictors of the TRA model affect intentions depends on a number of variables, including the type of behavior it is applied to [30]. Despite being very successful in explaining a variety of behaviors [31–34], the TRA model was extended by Ajzen [13] as it was later found to explain only volitional behaviors. Consequently, the TRA model was extended by Ajzen [13], and an additional construct, named "perceived behavioral control" (or PBC), was added to the original TRA model, and now it is called the "Theory of planned behavior" or TPB model.

Ajzen [13] defined PBC as "people's perception of the ease or difficulty of performing the behavior of interest" (p. 183). Thus, as per the TPB model, along with attitude and SNs, intentions are hypothesized to be determined by PBC, i.e., the degree to which an individual perceives a behavior to be in his control or not. The TPB model postulates that PBC influences actual behavior through behavioral intention as a mediator, as well as directly from PBC to actual behavior [13]. However, the relative importance of these three constructs (ATT, SNs, and PBC) in determining and predicting behavioral intentions varies from behavior to behavior. Different research studies have produced different results for the three constructs (attitude, SNs, and PBC) depending on the nature of the behavior to be measured [35].

Subsequently, the TPB model was applied to explain a variety of behaviors, including soil conservation practices [36]; conservation of water in the lodging context [37]; food waste behavior [38,39]; solid waste separation behavior [40]; entrepreneurship [41,42]; technology adoption [43,44]; unethical behavior [45]. The TPB model has also been well-applied in the domain of green or sustainable consumption to explain the consumer's intention to purchase and use environmentally sustainable products [15,46–48] and the consumption of organic products [49,50]. Therefore, the following hypotheses are proposed:

**H1:** *Attitude towards green products (ATT) positively influence green purchase intention (GPI).*

**H2:** *Subjective norms (SNs) positively influence green purchase intention (GPI).*

**H3:** *Perceived behavioral control (PBC) positively influences green purchase intention (GPI).*

Whilst applying the TPB model to explain pro-environmental behavior and sustainable consumption, modifications have been proposed and context-specific variables have been added to the original TPB model [51–54]. Consumers' pro-environmental behavior has been found to be influenced by a number of factors, including environmental concern and knowledge [55], moral and ethical obligation [56], and values and religiosity [57,58]. Thus, the present paper proposes a modification of the TPB model by including two more constructs, ethical obligation and green self-identity, to better explain the consumer's behavior towards green electronic products (see Figure 1).

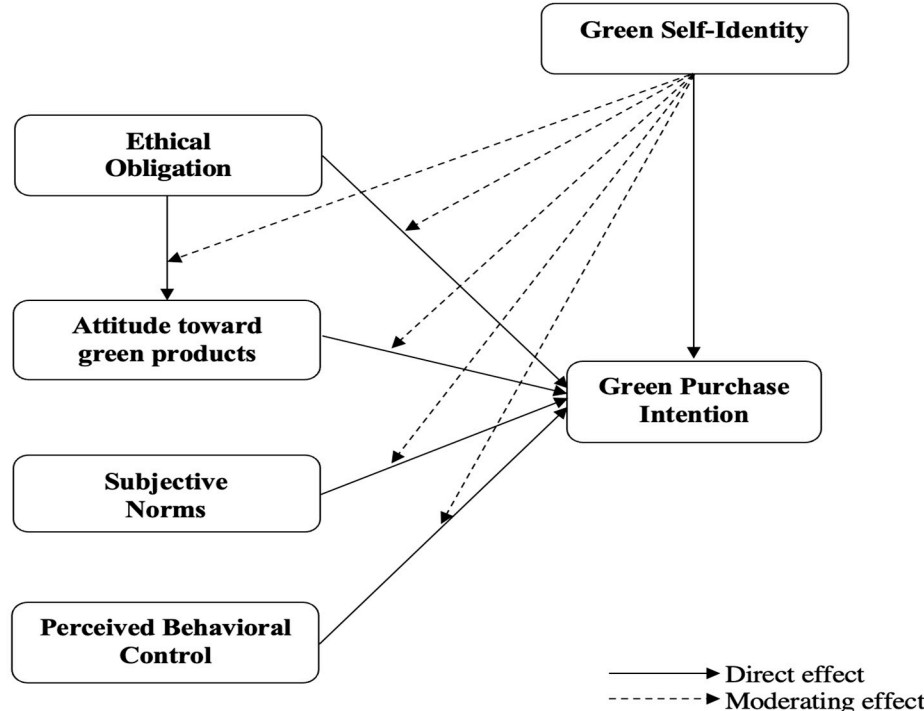

**Figure 1.** Proposed model.

*2.1. Ethical Obligation (EO)*

Ethical obligation is defined as "an individual's internalized ethical rules which reflect personal beliefs about appropriate behavior" [59] (p. 287). The extant literature suggests that ecologically conscious consumers tend to purchase environmentally friendly products that cause less harm to the environment [60]. However, their intention to purchase environment-friendly products is highly guided by psychological factors [61], such as a perceived moral norms [55,56] or ethical obligation [24] to contribute towards the social cause. Ethical obligation has been a crucial determinant of an individual's ethical behavior. Yoon [62] reported the significant influence of moral obligation on the intention to commit digital piracy. In addition, Shaw et al. [26] revealed that ethical consumers embrace a strong feeling of obligation to others, which impacts their purchase choices. Such feelings show an individual's internalized ethical rules and reflect their personal beliefs about wrong or right [56]. The majority of the studies indicated that ethical obligation exhibits a direct positive influence on purchase intention [26,59,63]. A few other studies, such as that by Oh and Yoon [64], demonstrated that ethical obligation exhibits an indirect effect on purchase intention through attitude. In addition, Arli et al. [65] conceptualized the perceived readiness to be green as a mediator between ethical obligation and purchase intention. Thus, the following hypotheses are proposed:

**H4:** *Ethical obligation (EO) positively influences green purchase intention (GPI).*

**H5a:** *Ethical obligation (EO) positively influences attitude towards green products (ATT).*

**H5b:** *Attitude towards green products (ATT) mediates the effect of Ethical obligation (EO) on green purchase intention (GPI).*

*2.2. Green Self-Identity (GSI)*

Green self-identity is the extension of the concept of self-identity [66] in the context of the consumption of green or sustainable products. Self-identity is defined as "the salient part of an actor's self which relates to a particular behavior" [67] (p. 1444). The notion of self-identity becomes very vital in explaining behavior in a different context as individuals relate themselves with said behavior. Thoits and Virshup [68] described self-identity in three distinctive aspects, i.e., "personal identity", "role identity", and "social identity". Personal identity refers to the tendency of a person to self-define themselves as a "unique and idiosyncratic character" [69] (p. 4). Whereas, in the case of role identity, an individual associates themselves with performing a particular role. In addition, social identity refers to the tendency of a person to associate themselves with a particular social group. Self-identity has been reported to be a crucial antecedent of behavior across various contexts, such as travel behavior [70], ethical consumption [59], physical activity [71], water drinking behavior [72], and entrepreneurial intentions [73].

Derived from the self-congruity theory [74], green self-identity indicates that the "individuals who perceive themselves as green consumers may consider purchasing eco-friendly products because these items satisfy their self-definitional needs, and they gain personal satisfaction from it" [27] (p. 193). Green identity has been reported as a major determinant of consumers' intention to purchase environmentally sustainable products. The extant literature indicates that the majority of the previous studies conceptualized green self-identity as a direct predictor of green purchase intention [25,75–78], sustainable consumption behavior [79], or pro-environmental behavior [80–82]. However, several studies have also demonstrated that green self-identity exhibits direct, as well as indirect, effects on purchase intention through various context-specific variables [83–85].

Thus, drawing conclusions from the above discussion, the following hypothesis has been proposed:

**H6:** *Green self-identity positively influences green purchase intention.*

Additionally, a few other studies have also conceptualized green self-identity as a moderator. For example, Agnihotri et al. [86] reported that green self-identity moderates the effect of satisfaction on revisit intention in green restaurants. Moreover, consumers with green self-identity were found to be less worried about service failures and willing to visit the hotel again in the future. Similarly, Neves and Oliveira [87] also considered green self-efficacy as a moderator while explaining behavior change in the context of energy-efficient heating appliances. In addition, Li et al. [88] and Carfora et al. [89] also examined the moderating effect of green self-efficacy while extending the TPB model to explain green purchase behavior. Thus, the current study also aims to investigate the moderating effect of green self-identity and proposes the following hypotheses:

**H6a:** *GREEN self-identity (GSI) moderates the effect of attitude (ATT) on green purchase intention (GPI).*

**H6b:** *Green self-identity (GSI) moderates the effect of subjective norms (SNs) on green purchase intention (GPI).*

**H6c:** *Green self-identity (GSI) moderates the effect of perceived behavioral control (PBC) on green purchase intention (GPI).*

**H6d:** *Green self-identity (GSI) moderates the effect of ethical obligation (EO) on green purchase intention (GPI).*

**H6e:** *Green self-identity (GSI) moderates the effect of ethical obligation (EO) on attitude towards green products (ATT).*

### 3. Materials and Methods

A cross-sectional research design with a quantitative approach was used in the present study. A sample of 386 respondents were chosen using a non-probability purposive sampling method among respondents residing in a few major cities of the northern-central part of India. Approximately 20–25% of India's population live in the northern part of India. Educated persons (above 18 years old) were contacted to collect the data as they are relatively more aware of environmental issues. All the respondents participated in the survey voluntarily. They were categorically informed about the purpose of the study and were also ensured that their information would be used only for academic purposes. The data for the study were collected using a structured questionnaire, which was developed from the existing literature, using a five-point Likert scale (where "1 = strongly disagree, 2 = disagree, 3 = neutral, 4 = agree and 5 = strongly agree"). Three major determinants of the TPB model, i.e., "Attitude," "Subjective norms, "and "Perceived behavioral control" (3-items each) were adapted from [13], [90], and [17], respectively. In addition, 3-item scales to measure "Ethical obligation" and "Green self-identity" were taken from [59] and [78], respectively.

Moreover, structural equation modelling using Amos version 22.0 was applied to analyze the data collected from the respondents. The demographic profile of the respondents is shown in Table 1. Out of the total 386 respondents, 68.4% were male, whereas the remaining 31.6% were female, which shows that the data of female respondents were approximately one-third of the total sample. The majority of the respondents (81.4%) were young (up to 30 years), and 11.9% were between 30 and 40 years old. Only a small fraction of the sample respondents (6.7%) were above the age of 40 years. The majority of the respondents were either graduates (59%) or postgraduates and above (35%). Looking at the occupational status of the respondents, one-fourth (22%) of the sample were employed, and the majority (70.4%) were students (67.1%), whereas the remaining (7.6%) were either unemployed, self-employed, retired, or a homemaker. As the majority of the respondents were students, they had no income (62.4%) and depend upon their families for their financial needs. About 13.7% of the respondents received an annual income between Rs. 2.5 Lakhs and Rs. 5 Lakhs (which is approximately 3000–6000 USD) and 9.1% between Rs. 5 Lakhs and Rs. 10 Lakhs (approximately between 6000–12,000 USD). Only a very small part of the sample (2.8%) had an annual income above Rs. 10 Lakhs (more than 12,000 USD), whereas 11.9% of the respondents did not reveal their annual income. Thus, the sample mostly represents young and educated Indian middle-class consumers.

**Table 1.** Demographic profile of respondents.

| Variable Name | Categories | Frequency | Percentage |
|---|---|---|---|
| Gender | Male | 264 | 68.4 |
|  | Female | 122 | 31.6 |
| Age (in years) | Up to 20 | 42 | 10.9 |
|  | 20–30 | 272 | 70.5 |
|  | 30–40 | 46 | 11.9 |
|  | Above 40 | 26 | 6.7 |
| Highest educational qualification | 12th or intermediate | 23 | 6.0 |
|  | Graduation | 228 | 59.0 |
|  | Post-Graduation and higher | 135 | 35.0 |

**Table 1.** *Cont.*

| Variable Name | Categories | Frequency | Percentage |
|---|---|---|---|
| Occupational status | Student | 266 | 70.4 |
| | Employed | 83 | 22.0 |
| | Others (Self-employed, unemployed, Homemaker, retired) | 29 | 7.6 |
| | Dependent on family income | 241 | 62.4 |
| Annual income (in Rs.) | Below 2.5 Lakhs | 0 | 0 |
| | 2.5–5 Lakhs | 53 | 13.7 |
| | 5–10 Lakhs | 35 | 9.1 |
| | More than 10 Lakhs | 11 | 2.8 |
| | Prefer not to say | 46 | 11.9 |

## 4. Results

As suggested by Anderson and Gerbing [91], the assessment of the proposed research model was conducted following a two-step approach. In the first step, the measurement model specifying the relationships between the constructs (latent variables) and the corresponding observed variables (items/statements) were examined, and the goodness of fit of the measurement model was assessed (also known as confirmatory factor analysis) using chi-square statistics. The measurement model produced an acceptable model fit with CMIN/df = 3.179 ($p$ = 0.00). A significant probability value is expected as test statistics are prone to a large sample size [92]. As chi-square statistics are sensitive to the sample size, various other model fit indices have been developed by researchers to ascertain the goodness of fit of the measurement model [93]. There are three types of model fit indices: absolute, incremental, and parsimony fit indices. Hair et al. [93] suggested reporting at least one from each type of indices to check the model fit. Table 2 shows that all the fit indices are as per the acceptable threshold value suggested by Browne and Cudek [94], for example, the absolute fit indices: CMIN/df = 3.719 (<3), GFI = 0.899 (>0.9), and RMSEA = 0.075 (<0.08); and the incremental fit indices: CFI = 0.946 (>0.9) and TLI = 0.928 (>0.9), and Parsimony fit index AGFI = 0.860 (>0.80) are within the acceptable range.

**Table 2.** Measurement model fit.

| S. No. | Fit Indices | Model Values | Recommended Threshold |
|---|---|---|---|
| 1. | CMIN | 435.465 | – |
| 2. | df | 137 | – |
| 3. | $p$ value | 0.000 | A significant $p$-value is expected |
| 4. | CMIN/df | 3.179 | <3 |
| 5. | GFI | 0.899 | >0.9 |
| 6. | AGFI | 0.860 | >0.8 |
| 7. | TLI | 0.928 | >0.9 |
| 8 | CFI | 0.946 | >0.9 |
| 9. | RMSEA | 0.075 | <0.1 (preferably less than 0.08) |
| 10 | SRMR | 0.039 | <0.1 (preferably less than 0.08) |

### 4.1. Reliability

Reliability is a measure of internal consistency. It refers to the extent to which the observed variables of a construct are internally consistent to one another. It is also an indicator of convergent validity [93]. One of the commonly accepted statistical measures of assessing reliability is Cronbach's alpha [95]. However, other measures of reliability are also available, which do not produce "dramatically different reliability estimates" [93] (p. 680). Composite reliability is another popular measure of internal consistency, which is generally used with structural equation modeling and is assumed to produce more accurate reliability estimates than Cronbach's alpha [96,97]. For a measure to be internally consistent, the values of the composite reliability should be more than 0.7 [98]. The values

of the composite reliability (given in Table 3) indicate that these values are more than the acceptable threshold of 0.7. Thus, the results confirm that all the measures of the study are internally consistent.

**Table 3.** Confirmatory factor analysis (Factor loading, composite reliability and AVE).

| S. No. | Constructs/Items | Factor Loadings | Composite Reliability | AVE |
|---|---|---|---|---|
| | Attitude towards green products (ATT) | | | |
| 1. | "I think using green products is wise." | 0.774 | | |
| 2. | "I think using green products is good." | 0.878 | 0.812 | 0.685 |
| 3. | "I think using green products is beneficial." | 0.877 | | |
| | Subjective norms (SNs) | | | |
| 4. | "My family would support my decision to purchase green products." | 0.872 | | |
| 5. | "My friends would support my decision to purchase green products." | 0.885 | 0.913 | 0.777 |
| 6. | "My colleagues would support my decision to purchase green products." | 0.887 | | |
| | Perceived Behavioral Control (PBC) | | | |
| 7. | "Decision to purchase green products is entirely up to me." | 0.739 | | |
| 8. | "If I want, I can easily purchase green products." | 0.852 | 0.874 | 0.699 |
| 9. | "I have resources, time and opportunity to purchase green products" | 0.908 | | |
| | Ethical Obligation (EO) | | | |
| 10. | "I take responsibility for ethical obligation of consumption." | 0.917 | | |
| 11. | "I take responsibility for the support of ethical consumption." | 0.724 | 0.859 | 0.672 |
| 12. | "I do the public good through consumption as a societal member." | 0.806 | | |
| | Green Self-Identity (GSI) | | | |
| 13. | "Supporting environmental protection makes me feel that I'm an environmentally responsible person." | 0.829 | | |
| 14. | "I feel proud of being a green person." | 0.835 | 0.839 | 0.636 |
| 15. | "Supporting environmental protection makes me feel meaningful." | 0.723 | | |
| | Green purchase intentions (GPI) | | | |
| 16. | "I would like to use green products." | 0.871 | | |
| 17. | "I would buy green products if I happen to see them in a store." | 0.872 | 0.915 | 0.729 |
| 18. | "I would actively seek out green products in a store in order to purchase it." | 0.846 | | |
| 19. | "I would patronize and recommend the use of green products." | 0.825 | | |

### 4.2. Convergent Validity

Convergent validity refers to how well the observed variables converge to the corresponding constructs [93]. Convergent validity can be assessed by either standardized factor loading or average variance extracted (AVE). As a rule of thumb, a standardized factor loading of more than 0.7 indicates that the measure has convergent validity. As shown in Table 3, the standardized factor loadings range between 0.723 and 0.917 and are more than the accepted threshold of 0.7. However, a more statistically accurate measure of convergent validity is the estimation of the AVE values. As shown in Table 3, the AVE value for each construct is more than the critical value of 0.5 [99], which confirms the assumption of the convergent validity.

### 4.3. Discriminant Validity

Discriminant validity is the measure of the extent to which two constructs are truly different [93]. It indicates that the latent constructs, as conceptualized in the research model, are unique and different from the other latent constructs. Two constructs are said to be different from one another if the $\sqrt{AVE}$ for those constructs is greater than the correlation between the constructs [99]. Table 4 provides the estimation of the discriminant validity. As is evident from these results (Table 4), the $\sqrt{AVE}$ for each construct is more than the inter-construct correlations, which confirms that the constructs possess discriminant validity.

**Table 4.** Discriminant validity.

|  | GPI | ATT | SNs | PBC | EO | GSI |
|---|---|---|---|---|---|---|
| GPI | 0.854 |  |  |  |  |  |
| ATT | 0.651 | 0.828 * |  |  |  |  |
| SNs | 0.590 | 0.444 ** | 0.881 |  |  |  |
| PBC | 0.720 | 0.491 | 0.572 | 0.836 |  |  |
| EO | 0.419 | 0.320 | 0.312 | 0.352 | 0.819 |  |
| GSI | 0.537 | 0.257 | 0.666 | 0.472 | 0.299 | 0.797 |

Note: * square root of AVE. ** inter-construct correlation.

### 4.4. Structural Model

After operationalizing the study constructs as the measurement mode in the first step, the causal relationships among the constructs were examined in the second step as the structural model. The results of the path analysis (i.e., structural model) are detailed in Table 5. The results indicate that among the three predictors of purchase intention in the TPB model, only two, i.e., ATT (estimate = 0.435; $p < 0.001$) and PBC (estimate = 0.389; $p < 0.001$), were found to exhibit a significant influence on GPI, while SNs (estimate = $-0.012$; $p > 0.001$) had no significant influence on GPI. In addition, of the two constructs added to the TPB model, only GSI (estimate = 0.194; $p < 0.001$) was reported to be positively related to GPI. In contrast, ethical obligation (estimate = 0.047; $p > 0.001$) exhibited no significant influence on GPI. However, ethical obligation was reported to exhibit a significant influence on attitude toward green products (estimate = 0.399; $p < 0.001$).

**Table 5.** Direct and moderating (interaction) effects.

|  |  |  | Unstandardized Estimate | SE | CR | $p$ | Standardized Estimates |
|---|---|---|---|---|---|---|---|
| EO | ---> | ATT | 0.399 | 0.050 | 8.024 | *** | 0.399 |
| ATT | ---> | GPI | 0.435 | 0.026 | 16.945 | *** | 0.475 |
| SNs | ---> | GPI | $-0.012$ | 0.039 | $-0.315$ | 0.753 | $-0.014$ |
| PBC | ---> | GPI | 0.389 | 0.032 | 12.099 | *** | 0.424 |
| EO | ---> | GPI | 0.047 | 0.029 | 1.628 | 0.103 | 0.052 |
| GSI | —> | GPI | 0.194 | 0.038 | 5.031 | *** | 0.211 |
| GSI × ATT | ---> | GPI | 0.084 | 0.024 | 3.586 | *** | 0.105 |
| GSI × SNs | ---> | GPI | $-0.036$ | 0.029 | $-1.254$ | 0.21 | $-0.067$ |
| GSI × PBC | ---> | GPI | $-0.021$ | 0.032 | $-0.637$ | 0.524 | $-0.039$ |
| GSI × EO | ---> | GPI | $-0.034$ | 0.023 | $-1.47$ | 0.142 | $-0.067$ |
| GSI × EO | ---> | ATT | 0.077 | 0.028 | 2.802 | 0.005 | 0.139 |

Note: *** $p < 0.001$.

Additionally, the study also hypothesized that the effect of ATT, SNs, PBC, and EO on GPI, and of EO on ATT, will be positively moderated by GSI. The results (Table 5) demonstrated that GSI only moderated the effect of attitude on GPI (interaction effect = 0.084; $p < 0.001$), while the moderating effects of GSI on the other three variables, i.e., SNs (interaction effect = $-0.036$; $p > 0.001$), PBC (interaction effect = $-0.021$; $p > 0.001$) and EO (interaction effect = $-0.034$; $p > 0.001$), were not significant. However, the effect of EO on ATT was found to be positively moderated by GSI (interaction effect = $-0.077$; $p > 0.001$). Finally, ATT emerged as the most important predictor of GPI, followed by PBC and GSI. In addition, these predictors were found to explain 75.6% of the total variance in GPI.

### 4.5. Mediation Analysis

Along with estimating the direct path, as conceptualized in the research model, mediation analysis was also conducted to estimate the indirect effect by applying a bootstrapping procedure [100] with 2000 sub-samples and a 95% bias-corrected confidence interval. The results of the mediation analysis, i.e., the direct, indirect, and total effects, are detailed in Table 6. In the present mediation model, consumers' attitude towards environmentally

sustainable products was conceptualized as the mediator between ethical obligation and green purchase intentions. The results showed that the standardized indirect effect of EO on GPI (through ATT) was 0.190 (lower bound = 0.128, upper bound = 0.260; $p = 0.000 < 0.001$). The significant indirect effect confirmed that attitude mediates the path between EO and GPI. Moreover, the standardized direct effect between EO and GPI (effect = 0.052, lower bound = −0.018, upper bound = 0.123; $p = 0.157 > 0.001$) was found to be non-significant, indicating that the relationship between EO and GPI was fully mediated by ATT.

**Table 6.** Mediation analysis (Direct, indirect, and total effect).

| Standardized Indirect Effect | | | | Standardized Direct Effect | | | | Total Effect | |
|---|---|---|---|---|---|---|---|---|---|
| **Effect** | **Lower Bound** | **Upper Bound** | **Two-Tailed Sig.** | **Effect** | **Lower Bound** | **Upper Bound** | **Two-Tailed Sig.** | **Effect** | **Two-Tailed Sig.** |
| 0.190 | 0.128 | 0.260 | 0.000 | 0.052 | −0.018 | 0.123 | 0.157 | 0.241 | 0.000 |

## 5. Discussion

The consumption of environment-friendly products is largely influenced by a person's compliance with ethical and moral norms and their willingness to contribute to the cause of saving the environment for the future generation. Consumers who are concerned about environmental degradation have a greater appreciation for the products that cause less harm to the environment and ecology (also known as green or environmentally sustainable products) compared to their traditional counterparts [101]. Consumers' inclination to purchase green products is also the manifestation of their ethical obligation towards society and the environment. In addition, such consumers also purchase and consume green products to fulfill their personal goals of distinguishing themselves as socially and environmentally responsible citizens. Thus, the present study proposed to include two constructs, i.e., ethical obligation and green self-identity, to investigate green purchase intentions, utilizing the TPB model as the basic framework of the study.

The results of the current study (see Figure 2) demonstrated that attitude towards green products and perceived behavioral control were found to be the major predictors of green purchase intention, whereas subjective norms were found to exhibit no significant impact on green purchase intentions. Attitude, subjective norms, and perceived behavioral control are the three major predictors of behavior, as per the TPB model [13]. However, the results of the current study demonstrate that only two of the three predictors were found to be significant in influencing green purchase intention.

Consumers' attitude towards the products is the primary determinant of their intention to purchase the products. The present study demonstrates that consumers' favorable disposition towards green products shapes their intention towards purchasing these green products. Moreover, perceived behavioral control was reported to be another important antecedent of green purchase behavior. Purchasing green products (which are relatively costlier than their traditional counterparts) is non-volitional and subjected to numerous restrictions. Therefore, an individual's perceived control over their behavior indicates whether they will be inclined towards buying green products or not. These results are consistent with many previous studies that have reported attitude and perceived behavioral control as the main determinants of sustainable consumption behavior [14–16]. Several studies conducted in the domain of sustainable consumption have reported subjective norms to have either a reduced impact or no impact on purchase intention [18,19,33,43]. The findings of the present study corroborate the previous findings. Moreover, Sparks et al. [24] and Shaw et al. [26] argued that when consumers are guided by the notion of ethical obligation and self-identity, the role of others (i.e., subjective norms) becomes relatively less important while making purchase decisions.

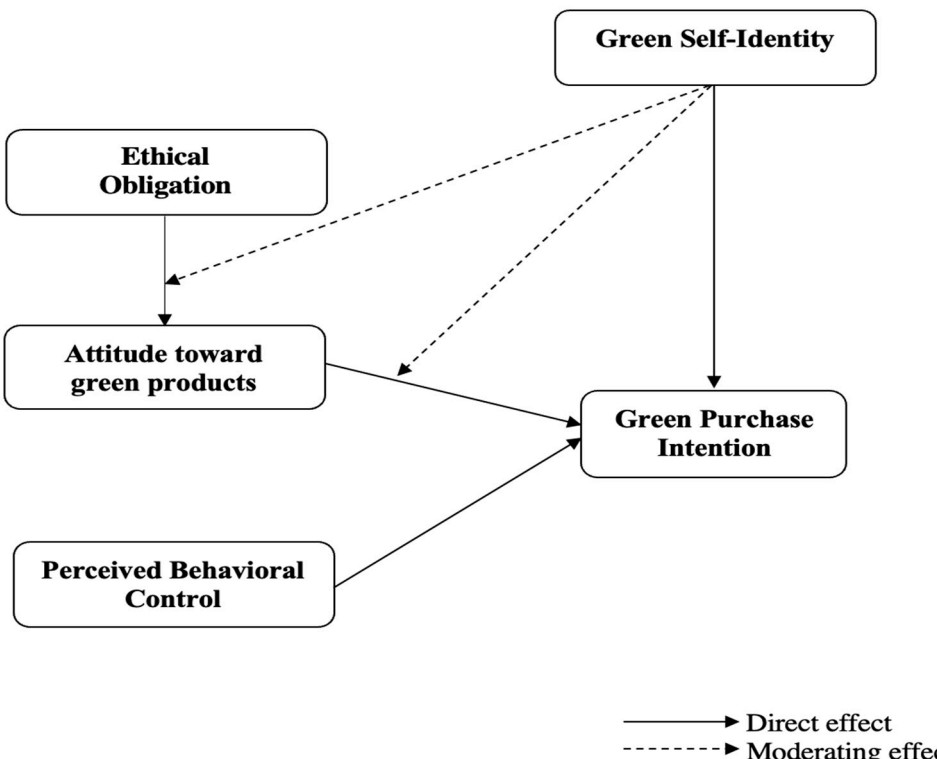

**Figure 2.** Final model.

Furthermore, of the two variables added to the TPB model, only green self-identity was found to be positively related to green purchase intentions. Ethical obligation, on the other hand, was found to have no significant influence on purchase intention. However, ethical obligation was found to be positively related to attitude towards green products. In addition, the results of the mediation analysis confirmed that the effect of ethical obligation on green purchase intention was fully mediated through attitude towards green products. Thus, consumers' feelings of ethical obligation will not directly stimulate favorable purchase intentions; rather, ethical obligations will strengthen consumers' attitude towards green products, which in turn will influence their purchase intentions. In a nutshell, the findings signify the important role of ethical obligation in stimulating green purchase intentions, which is in line with previous studies [26,59,64,75,77].

Additionally, the present study proposed green self-identity as a moderator and estimated the interaction effect of green self-identity with the remaining constructs of the TPB model. The results demonstrated that green self-identity positively moderated the influence of attitude on green purchase intentions, which indicates that for those who characterize themselves more as green consumers, the impact of attitude on their purchase intention will be greater compared to those who perceive themselves less as green consumers. Moreover, the findings also revealed that green self-identity also moderated the effect of ethical obligation on attitude. Thus, green self-identity strengthens the impact of ethical obligation on a person's attitude towards green products. These results establish green self-identity as a moderator and enrich the extant literature [27,75,84,85]. Thus, the present study successfully extends the TPB model to include ethical obligation and green self-identity while explaining green purchase behavior.

## 6. Implications

The results of the study provide implications that are theoretically meaningful. Furthermore, these results give practical implications for marketers and policymakers. Theoretically, the current study extends the TPB model to include ethical obligation and green self-identity, which are rarely investigated together. In addition, the study examines the

moderating role of green self-identity, which has rarely been explored in the previous literature. Thus, the present study contributes to the existing literature by filling this gap. In addition, as shown by the results, a favorable attitude towards environmentally sustainable products is the key to explaining and predicting consumers' intention to purchase these products. Thus, marketers should attempt to stimulate consumers' attitude by using promotional techniques. They should attempt to make the consumers aware of the benefits of using environmentally sustainable products and the harm caused by non-sustainable products with promotional campaigns. The government and NGOs should create awareness programs on the community level to create awareness among citizens about sustainable products. In addition, perceived behavioral control and green self-identity were found to be major predictors of the purchase intention towards sustainable and green products. Furthermore, ethical obligation was reported to exhibit a positive influence on attitude toward green products. Although subjective norms were found to have no influence on purchase intention, the role of social surroundings remains crucial. In fact, when people are self-aware and motivated toward social and environmental issues, the role of societal norms becomes relatively less influential in stimulating favorable purchase intentions. Therefore, the government and policymakers should attempt to create and promote the use of sustainable products on the community level. When people in one's surroundings use these products, others will also be motivated to purchase and use such products. Moreover, an individual's ethical and moral norms play a crucial role in shaping a favorable attitude toward sustainable products; therefore, attempts should be made to make people environmentally responsible. If a consumer is informed about environmental issues and is concerned about reducing carbon emission and saving and protecting the environment, they will be inclined towards purchasing such products. Nukkad natak and other such promotional tools should be used by the government and NGOs to make people aware of the environmental issues and the importance of saving the environment by avoiding the use of environmentally harmful products. In addition, such an initiative will enforce a sense of individual and collective responsibility towards the environment. Consumers should also be educated and persuaded about their key role in reducing carbon emissions. If consumers identify themselves as socially and environmentally responsible individuals, they will not only use sustainable products themselves, but also promote the use of such products in their social surroundings.

## 7. Conclusions, Limitations and Future Research

The current study focused on examining the role of ethical obligation and green self-efficacy in predicting green purchase intentions. The results of the study confirm the applicability and suitability of the theory of planned behavior in explaining consumers' intentions to purchase sustainable products. In addition, the study investigates the role of ethical obligation and green self-identity in stimulating favorable intentions to engage in sustainable consumption. The results showed that ethical obligation influences attitudes towards green products, while green self-identity positively influences green purchase intention. In addition, green self-identity was found to moderate the effects of attitude and ethical obligation on green purchase intentions.

Furthermore, as is the case in any other research study, the present study also includes a few limitations. The study focused mainly on consumers' intention to purchase sustainable products due to the cross-sectional research design. However, a gap exists between intention and actual purchase behavior, which may be addressed by future researchers using a longitudinal design. Consumer behavior is very dynamic in nature and is influenced by a number of sociodemographic and psychological factors. The present study only took into account two such factors, i.e., ethical obligation and green self-identity. Future studies may identify other important factors (willingness to pay a premium for green products, values, spirituality, religiosity) and investigate their role in sustainable consumption. Future studies may also focus on exploring the role of values such as the environmental, emotional, social, and economic value of purchasing green products.

**Author Contributions:** R.K. and K.K. conceived of the idea; R.S., R.K. and K.K. surveyed the literature and developed the conceptual framework and proposed the hypothesis; J.C.S. collected the responses from respondents and conducted the pilot study; R.K. and R.S. contributed to the analysis; R.S., R.K., J.C.S. and K.K. wrote the paper; G.S. and S.C. edited the original draft and administrated the project. All authors have read and agreed to the published version of the manuscript.

**Funding:** This research received no external funding.

**Institutional Review Board Statement:** Not applicable.

**Informed Consent Statement:** Not applicable.

**Data Availability Statement:** The data are available on request.

**Conflicts of Interest:** The authors declare no conflict of interest.

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
