# Peer review of "Modeling Environmentally Conscious Purchase Behavior: Examining the Role of Ethical Obligation and Green Self-Identity"

_sustainability, doi:10.3390/su15086426_

Round 1
Reviewer 1 Report
Dear Authors, thank you for an interesting manuscript. I think it will attract high interest of other researchers. However, before publishing, I have some suggestions for improvement.
Abstract. The abstract is adequate. The authors describe the research and provide main results. However, further research implications and the scientific value are not established.
Introduction. The relevance of the topic is presented quite clearly. However, more emphasis on previous research on green self-identity and ethical obligation must be provided. The choice of expanding TPB model with these two dimensions is not substantiated. Therefore, it is not clear whether the gap in literature exists. Try to demonstrate that these two dimensions deserve to be included into the models.
The research methods and pursued results are not presented. Also, the last paragraph in the Introduction section could be provided to explain what each section of the paper is about.
Theoretical framework and hypotheses. Theoretical framework is arranged logically and provides a substantial background for the hypotheses and the model. My only doubt is about Table 1. On the one hand, this table is very interesting and provides valuable information for the readers. On the other hand, its necessity in this particular article is doubtful. The table demonstrates extensions of the model having no relation to current research. So, either the table must be removed, either a proper explanation and substantiation must be provided.
Materials and methods. The authors provide information about the sample and the structure of the questionnaire clearly. However, as a journal is being read internationally, I suggest providing some internationally used currency as a reference, while describing income (e.g., US dollars or Euro: ).
Results. The results are presented clearly, and all the necessary indicators are calculated to substantiate the model. However, I would suggest adding a figure with the final (statistically verified) model.
Discussion and conclusions. Both discussion and conclusions are provided correctly, maybe future research guidelines could have been provided more clearly.
Author Response
Comments and Suggestions for Authors
Dear Authors, thank you for an interesting manuscript. I think it will attract high interest of other researchers. However, before publishing, I have some suggestions for improvement.
Abstract. The abstract is adequate. The authors describe the research and provide main results. However, further research implications and the scientific value are not established.
Response: We extend our sincere thanks for your valuable suggestions. We have modified abstract to include your suggestions (Please see lines 29-31)
Introduction. The relevance of the topic is presented quite clearly. However, more emphasis on previous research on green self-identity and ethical obligation must be provided. The choice of expanding TPB model with these two dimensions is not substantiated. Therefore, it is not clear whether the gap in literature exists. Try to demonstrate that these two dimensions deserve to be included into the models.
Response: We extend our gratitude for careful reading of the manuscript giving valuable suggestions. We have included above suggestions in the revised manuscript (Please see lines 82-100).
The research methods and pursued results are not presented. Also, the last paragraph in the Introduction section could be provided to explain what each section of the paper is about.
Response: We thank you for your valuable suggestions. We have now included a para on the outline of the manuscript (Please see lines 129-136).
Theoretical framework and hypotheses. Theoretical framework is arranged logically and provides a substantial background for the hypotheses and the model. My only doubt is about Table 1. On the one hand, this table is very interesting and provides valuable information for the readers. On the other hand, its necessity in this particular article is doubtful. The table demonstrates extensions of the model having no relation to current research. So, either the table must be removed, either a proper explanation and substantiation must be provided.
Response: We extend our gratitude for careful reading and pointing out these issues. We had prepared this table (Table 1) to highlight what context-specific variables have been added to the original TPB model while extending it in other domains. However, in-line with your valuable suggestion we have removed the table from the text.
Materials and methods. The authors provide information about the sample and the structure of the questionnaire clearly. However, as a journal is being read internationally, I suggest providing some internationally used currency as a reference, while describing income (e.g., US dollars or Euro: ).
Response: We extend our thanks for your valuable suggestions. We have converted annual income figures in US dollars (from Indian Rupees) as suggested to make it readable to the international readers (Please see lines 576-578).
Results. The results are presented clearly, and all the necessary indicators are calculated to substantiate the model. However, I would suggest adding a figure with the final (statistically verified) model.
Response: We extend our sincere thanks for your appreciations. As suggested, we have incorporated one more figure (Figure 2) which presents final statistically verified) model (Please see lines 784-791).
Discussion and conclusions. Both discussion and conclusions are provided correctly, maybe future research guidelines could have been provided more clearly.
Response: We are thankful for your feedback. Future guidelines have been further revised as suggested (Please see lines 859-861).

Reviewer 2 Report
Please find attached file.

Author Response
Topic: The topic is appropriate, coving all the important aspects of current research.
Response: We extend our gratitude for your positive feedback.
Abstract
- Abstract is well-written and presented.
Response: We convey our sincere thanks for your appreciation.
Keywords: Appropriate
Response: We are thankful to you for your positive comment.
Introduction
- Correct write 1oC as 1o It is creating a lot of mess.
Response: The above typing error has been dully corrected in the revised manuscript (Please see line 41).
- Authors have cited a lot of work on the variables (like see line 65 to 73) but haven’t
described the research gap which is an essential part of introduction section.
Response: We are thankful for your valuable feedback. We have highlighted the research gap in the revised manuscript (Please see lines 82-100).
- Never end any paragraph with reference(s).
Response: As suggested we have extended the paragraph and ended on the text rather than citation (Please see lines 66-67).
- Author should also answer the “why” related to the concepts of current research and their relationships. Otherwise, the research’s importance is negatively affected.
Response: Thank you for your feedback. We have given reasons of ‘why’ related to the concepts of the current research and their relationships (Please see lines 82-100).
- In the introduction section, normally the author provides detail about the background of the study, underpinning theories, gap(s) in the literature that this study will fill, the objectives of the study, and then provides a breakdown of the current manuscript. The author should revisit this section and incorporate the suggested changes.
Response: We are thankful for your valuable suggestions. We have rearranged the introduction section in the order suggested by you (Please see lines 35-136).
- It is also recommended to provide a brief introduction of the population of the study in this section also.
Response: Population of the study has been defined in the introduction section as suggested (Please see lines 102-103).
- While introducing any concept, it is important to also provide its definition.
Response: We extend our sincere thanks for your valuable suggestions. We have given definition of the concept while introducing them in the introduction section (Please see lines 84-91).
Theoretical framework and hypotheses
- Define the abbreviations on their first appearance like the reader don’t know what is SNs?
Response: As suggested Abbreviations have been defined at their first appearance (Please see line 143).
- Avoid using ‘etc.’ in academic writings. Provide 2-3 more examples and remove etc.
Response: We extend our sincere thanks for pointing out the above errors. The same have been duly rectified in the revised manuscript.
- Instead of explaining the whole theory, the focus must be on the hypothesis development and how this theory is explain that particular hypothesis or the relationship between the variable(s). All the other material is secondary.
Response: We are thankful for your feedback. As suggested we have shorten the background of the theory and focused on hypotheses development only.
- Start the literature review of any variable with its operational definition.
Response: As suggested literature review has been started with the definition of the variables (Please see lines 254-255).
- In all the hypotheses, remove the word ‘will’. Like H1 states “Attitude towards green products (ATT) will positively influence green purchase intention (GPI)”. It should be corrected as “Attitude towards green products (ATT) positively influence green purchase intention (GPI)”.
Response: As suggested the word ‘will’ from all the hypotheses have been removed (Please see lines 170-174; 271-275; 513-523).
Materials and Methods
- Why non-probability sampling method is utilized? With non-probability sampling technique, different tests are normally used.
Response: Thank you for your feedback. We have used non-probability sampling due the unavailability of the sampling frame.
- How you defined educated or middle class for the current study?
Response: We extend our sincere thanks for your comment. Educated refers to those who are at least graduate and middle class means those whose annual family income is more than 3000 USD.
- What was the total population of the study, write about it if even unknown.
Response: As suggested we given approximate estimate of the populate (Please see lines 551-553).
- What was the appropriate sample size and how it was determined?
Response: Sample size of the study was 386 responses. As per Hair et al. (2010), sample size should be 10-15 times the no of the items (to measure study constructs). In our study, there are 19 items measured on Likert scale and 15 times of this number (i.e. 19) comes out to be 285.
- How sample was selected?
Response: Thanks for your careful reading of the manuscript. Procedure of the sample selection has been detailed in the revised manuscript (Please see lines 551-553).
- In this section, normally detail of the following things is given:
- Research philosophy
- Research approach
- Research strategy
- Research method
- Time horizon
- Data collection and analysis
- Scale development
- Population and sample
- Software used
- Data collection
- Data preparation, editing, coding, entry, and cleaning
- And ethical considerations
Add the missing detail in this section as this is the most important part of the study.
Response: This section has been arranged in the above order as suggested (Please see lines 547-580).
Results
- What was the impact of controlled variables in the study? Not presented
- Refer all the table in text.
- Don’t end the section on the table.
Response: Above suggestions have been dully incorporated in the revised manuscript.
Discussion
- Discuss each hypothesis in detail. Provide data to support your hypothesis acceptance followed by theoretical support.
Response: We thank you for careful reading. We have discussed each hypotheses in discussion section one by one (Please see lines 741-760)
- Also discuss the underpinning theories with relationship to hypothesis. You have just given theory in the start and that the end.
Response: We are thankful for your valuable feedback. We have incorporate the discussion on theory in the discussion section now. (Please see lines 741-760)
- Add implication as a separate section. Here you have given a couple of implications in conclusion only. There should be a detailed implication (theoretical, practical as well as methodological).
Response: Thank you for your feedback. We have given detailed implications in a separate section in the revised manuscript (Please see lines 791-839).
- Link the theory within implications.
Response: We thank you for your comment. The same has been incorporated in the revised manuscript Please see lines 792-797).
